# Effect of Aerated Irrigation on the Growth and Rhizosphere Soil Fungal Community Structure of Greenhouse Grape Seedlings

**Huanhuan Zhang** [1,†], **Jinshan Xi** [1,†], **Qi Lv** [2], **Junwu Wang** [1], **Kun Yu** [1,*] and **Fengyun Zhao** [1,*]

[1] The Key Laboratory of Characteristics of Fruit and Vegetable Cultivation and Utilization of Germplasm Resources of the Xinjiang Production and Construction Corps, Shihezi University, Shihezi 832003, China

[2] Garden Science and Technology Branch, Xinjiang Agricultural Vocational Technical College, Changji 831100, China

[*] Correspondence: yukun@shzu.edu.cn (K.Y.); zhaofy@shzu.edu.cn (F.Z.)

[†] These authors contributed equally to this work.

**Abstract:** Conventional irrigation methods decrease greenhouse soil aeration, which leads to restricted root growth and reduced soil fungal abundance in greenhouse grapes. In this study, aerated irrigation equipment was used to investigate the effects of aerated irrigation on the biomass accumulation, root growth, and soil fungal community structure of grape seedlings. The results show that aerated irrigation significantly increased the root length, root surface area, root volume, and number of root tips by 38.5%, 32.1%, 62.1%, and 23.4%, respectively, at a soil depth of 20–40 cm ($p \leq 0.05$). The chao1 index and ACE index of fungi at different soil depths under aerated irrigation were higher than those without aerated treatment; aerated irrigation changed the relative abundance of dominant fungi in rhizosphere soil. At a soil depth of 20–40 cm, aerated irrigation increased the abundance of *Fusarium* by 42.2%. Aerated irrigation also contributed to the abundance of the beneficial fungal genera *Mortierella*, *Cladosporium*, and *Glomus*. At a soil depth of 0–20 cm, the abundance of *Mortierella* in the soil that received aerated treatment was 180.6% higher than in the control treatment. These findings suggest that aerated irrigation is a promising strategy for the promotion of grape root growth and biomass accumulation, and it can also increase the abundance of some beneficial fungi.

**Keywords:** aerated irrigation; root growth; beneficial fungal genera; fungal community structure; grape



## 1. Introduction

Subsurface drip irrigation is a widely used irrigation technology. Long-term subsurface drip irrigation has an impact on the soil's structure and hydraulic properties near the drippers, as well as limiting oxygen diffusion in the root zone of crops, thereby affecting the aerobic respiration of plant roots [1]. Such traditional irrigation techniques drive out the gas in the spaces between soil pores, thus reducing the soil air content between plant roots and causing low oxygen stress in plants [2,3]. Hypoxic stress is one of the important adversity factors affecting normal plant growth and development [4], which increases the reduction of toxic substances in the soil, imbalances the uptake of mineral elements, and disrupts the hormone metabolism in plants [5,6]. Aerated drip irrigation is a new type of irrigation that has emerged in recent years as a refinement and improvement of subsurface drip irrigation. Previous studies have shown that this irrigation method is an innovative water-saving irrigation technology with great development potential [7–9]. On the basis of traditional subsurface drip irrigation water-saving technology, aerated irrigation uses aeration equipment to dissolve air or other gases into irrigation water; once the water has been mixed with gas, it is then transported to the root zone of the crop through a drip irrigation pipeline [10]. Aerated irrigation can increase soil oxygen content and porosity, alleviate oxygen shortages during irrigation, and promote soil microbial and root respiration, improving root traits and biomass [11–14]. Aerated irrigation can improve

the ratio of solid, liquid, and gas phases in the soil, which can effectively regulate soil microbial activity, the effectiveness of soil nutrients and soil redox reactions, and other soil microenvironments [9]. In turn, it improves soil properties and increases soil productivity, thus enhancing root uptake and utilization of soil nutrients and water [15,16].

Previous studies on aerated irrigation have shown that aerated irrigation can promote plant growth, increase crop yield, and improve fruit quality [16–18]. Zhao et al. [19] foregrounded plant photosynthesis and growth in their study. Yuan et al. [20] reported that increasing oxygen levels in the root zones of crops had a positive impact, stimulating above-ground growth and increasing tomato yield increased by 8% on average. However, there are relatively few studies on the effects of subsurface drip irrigation and aerated irrigation on soil microbial diversity and community structure. Soil microorganisms are an important part of soil organisms; moreover, microbial diversity represents the stability of the microbial community and can objectively reflect the impact of soil ecological mechanisms on the community [21,22]. In an aeration experiment on melon crops, Li et al. [23] demonstrated that aerated irrigation had a significant effect on the quantity of microbes in the soil. Rhizosphere soil contains a large number of microorganisms such as fungi. Fungi are an important part of the soil–plant ecosystem, with the composition and diversity of fungal community partly determining plant growth [24,25]. Beneficial fungi abound in rhizosphere soil. These rhizosphere microorganisms can inhibit pathogenic fungi and thereby promote plant growth in a number of ways, including nutrient competition, antagonism, and the induction of systemic resistance [26]. *Cladosporium* has certain biocontrol effects: for example, improving the water utilization rate of plants under drought stress [27]. *Glomeromycota* is a type of arbuscular mycorrhizal fungi that can form mycorrhizal symbionts with most plant roots to promote plant growth. *Chaetomium* can be used as a biocontrol bacterium and is able to control plant pathogens [28]. Meanwhile, *Fusarium* causes root rot in plants. Root rot reduces a plant's ability to absorb water, which in turn impacts the normal function of the root system [29]. Therefore, the related research on the effect of aerated irrigation on the structure of the soil fungal community has both theoretical and practical significance for verifying the effectiveness and sustainability of rhizosphere aeration.

At present, China has an area of over $1.33 \times 105$ hm$^2$ used for cultivating facility grapes, making it the largest facility grape producer in the world [30]. In facility production, factors such as high intensification, over-irrigation, agricultural machinery rolling, excessive fertilization, and a lack of intertillage all lead to soil compaction and hypoxic stress in the root zone, thereby limiting the improvement of grape yield and quality [31]. The "water, fertilizer, and air" integrated subsurface drip irrigation technique is a new water-saving drip irrigation method, developed by combining drip irrigation with the "hole storage and fertilizer technology" proposed by academy member Shu Huairui [32]. Aerated irrigation can effectively solve the problems of root uplifting caused by surface drip irrigation and by the small infiltration range of the root zone of traditional subsurface drip irrigation. The technique is suitable for the production of multi-year fruit trees [33]. This technology has been proven to have certain advantages in promoting the deep growth of the roots of perennial fruit trees, as well as in water saving and fruit yield improvement [34,35]. However, there is a lack of systematic studies on the effects of aeration irrigation on grape growth and development and rhizosphere soil fungal community. In this study, the effects of rhizosphere aeration during subsurface drip irrigation on grape root growth and the soil fungal community structure were investigated using a greenhouse experiment, with the three-year-old 'Red Globe' grape as the test material. This study provides a theoretical basis for scientific soil aeration, as well as the optimization of aeration techniques, using subsurface drip irrigation with tanks.

## 2. Materials and Methods

### 2.1. Experimental Materials and Design

The test was carried out in a solar greenhouse (44°26′ N, 85°95′ E) at the Experimental Station of the Agricultural College of Shihezi University, Shihezi city, Xinjiang Province, from April to October 2016. During the test, the temperature of the greenhouse was 17–33 °C and the relative humidity was 60–80%.

The test was conducted in hard polyvinyl chloride planting boxes with a length and width of 40 cm and a height of 60 cm (Figure 1). Two treatments, i.e., the aerated treatment (aerated subsurface drip irrigation) and the control treatment (CK) were set in our test, with the 'Red Globe' grape as the test variety. Before the seedlings were transplanted, the assembled planting boxes were placed into a square soil pit 60 cm in depth. The spacing of the planting boxes was 20 cm. There was a total of 3 plots, with 9 boxes in each plot. The soil was obtained from the 0–20 cm topsoil in the vineyard of the Experimental Station of the Agricultural College of Shihezi University. The soil was gray desert soil (clay soil), which was sieved via a 0.425 mm mesh sieve. The pH value of the soil was 6.56. The soil contained 13.61 g kg$^{-1}$ organic matter, 1.22 g kg$^{-1}$ total N, 2.63 mg kg$^{-1}$ ammonium N, 2.01 mg kg$^{-1}$ nitrate N, 43.6 mg kg$^{-1}$ available P, and 305 mg kg$^{-1}$ available K. The soil bulk density was 1.40 g cm$^{-3}$. On May 6, 2016, the 3-year-old 'Red Globe' grape seedlings with uniform thickness and consistent growth were transplanted in the center of the planting box, and they were subjected to subsurface drip irrigation using the tank technique. The amount of irrigated water was the same for each plant. One month after planting, the grapevines were aerated using a self-designed water–fertilizer–air integration system that was based on subsurface drip irrigation with tanks for aerated treatment (Figure 1). This device was powered by electricity generated by solar panels. The formula for the injected air volume is V = 0.001 SH(1 − $\rho_b/\rho_s$)n [36], where V is the volume of air injected each time, L; S is the surface area of the polyvinyl chloride planting box, cm$^2$; H is the height of the polyvinyl chloride box, 60 cm; $\rho_b$ is the soil bulk density, 1.40 g cm$^{-1}$; $\rho_s$ is the soil density, 2.65 g cm$^{-3}$; and n is the number of aeration boxes [9]. The minimum air injection volume for a single test was calculated to be 520.3 L. Considering the effusion effect of the air stored in underground tanks and the results of the trial test, the aeration frequency was set at 20 min per day from 9:00 to 9:20 a.m. Each aeration lasted 20 min. The aeration tank was a polyvinyl chloride pipe with a diameter of 8.5 cm and a height of 10 cm. The upper part of the aeration tank was sealed, except for the air inlet holes, and 0.3 cm diameter micropores were evenly distributed across the lower part. The aeration tank was placed 5 cm away from the plant at a buried depth of 30 cm. A switch was installed between the main pipe and the branch pipe to control the air injection.

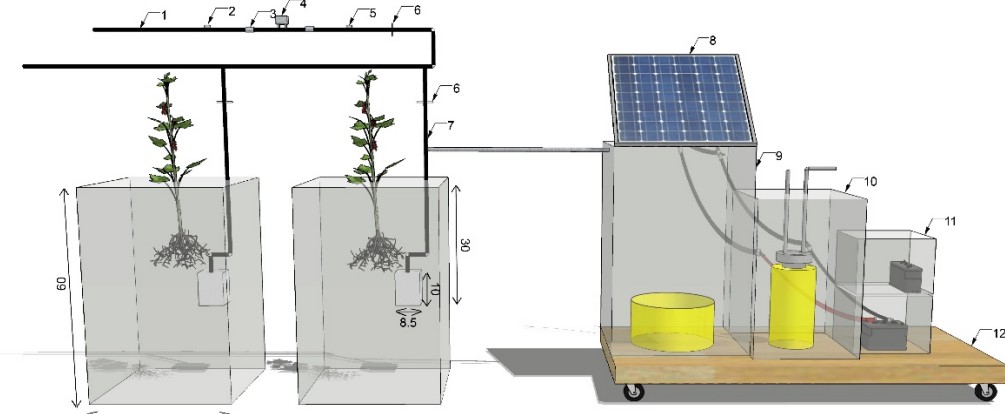

**Figure 1.** Aeration system for subsurface drip irrigation with tanks. (1) Main pipe, (2) water pump, (3) filter, (4) aeration system, (5) water meter, (6) switch, (7) branch pipe, (8) solar panel, (9) gas storage tank, (10) air pressure balancing tank, (11) battery control box, (12) supporting plate.

## 2.2. Soil Sampling

Soil samples were collected after 90 days of subsurface drip irrigation and aerated treatment. The above-ground parts of the grape seedlings were pruned and cut off, and the root drill method was used for sampling. Four points were evenly selected in each planting box; the points had a horizontal distance of 5 cm from the center of the aeration tank. Then, 100 g soil samples were taken at depths of 0–20, 20–40, and 40–60 cm below the four points, and the soil samples of one planting box were mixed with four soil samples of the same layer; this was repeated three times. The soil samples were immediately sieved through a 2 mm mesh, mixed well, and stored in a −80 °C refrigerator until 16 S high-throughput genome sequencing sampling was conducted.

## 2.3. Determination of Growth Index and Plant Biomass

After 57 d of aerated treatment, 3 labeled grape seedlings were selected for destructive sampling from each treatment, and the whole plant was divided into 6 parts: leaf, shoots, old branches, new branches, fine roots (<2 mm), and thick roots (>2 mm). The roots were sampled using the stratified sampling method [37]. The roots and soil were placed on a 100-mesh steel sieve for rinsing using a treatment sequence of clean water, detergent, clean water, 1% hydrochloric acid, and deionized water; this was repeated 3 times. The rinsed roots were placed in a 105 °C oven for 15 min and then dried at 75 °C to a constant weight and weighed. After that, the samples were smashed by an electric mill, sieved through a 150-mesh sieve, and bagged for later use.

## 2.4. Determination of Root Morphological Index

Three plants were selected from each treatment 90 d after aerated irrigation. The root index at every 20 cm was regarded as one layer (0–20, 20–40, and 40–60 cm). All the soil in each layer was dug out separately, and the roots were collected and washed quickly. These roots were scanned with a root scanner. The scanned root images were analyzed by WinRHIZO (Canada) [38] to obtain the effective root surface area ($cm^2$), effective root volume ($cm^3$), root length (cm), number of root tips, and root diameter (mm). After scanning, the roots were dried at 80 °C to constant weight and weighed to obtain root weight.

## 2.5. Soil DNA Extraction and PCR Amplification

The soil's total DNA was extracted from 0.5 g of soil using the FastDNA Spin Kit (MP Biomedicals, Cleveland, USA). After DNA extraction, the purity and concentration of the DNA were determined by agarose electrophoresis. The DNA was diluted to 1 ng $\mu L^{-1}$ in sterilized ultra-pure water and stored at −80 °C for later use. ITS5-1737F (GGAAG-TAAAAGTCGTAACAAGG) and ITS2-2043R (GCTGCGTTCTTCATCGATGC) [37] were used as primers to amplify the fungi in the ITS region. The amplification system was 20 μL (4 μL 5 × Fastpfu Buffer, 2 μL 2.5 mmol $L^{-1}$ dNTP, 0.8 μL 1737F, 0.8 μL 2043R, 0.4 μL Fastpfu Polymerase, 0.2 μL BSA, 10 ng DNA). PCR was performed under the following conditions: 3 min of denaturation at 95 °C, followed by 30 cycles, 30 s of denaturation at 95 °C, 30 s of annealing at 55 °C, 30 s of extension at 72 °C, and a final extension step at 72 °C for 5 min. After amplification, 3 μL of PCR products were detected using 2% agarose gel electrophoresis. Alpha diversity was evaluated using the QIIME suite of programs, including the Shannon and Simpson diversity indices and the Chao1 and ACE richness indices. Principal component analysis (PCA) was used to analyze the differences in the community structure of the different groups.

## 2.6. Bioinformatics Analysis

According to the barcode sequence and PCR amplification primer sequence, the data of each sample were split from the downstream data. The sequence of the barcode and primer was intercepted, and FLASH, v1.2.11 [39] was used to splice the sequence of each sample to obtain high-quality original data. For high-quality sequence data, the quality

control process was carried out by referring to QIIME [40], and chimeric sequences were further removed to obtain the final effective data. UPARSE software, v11.0.667 (Tiburon, CA, USA) [41] was used to cluster the final effective sequences, and OTUs (operational taxonomic units) were obtained by default with 97% sequence similarity. On the basis of the Greengenes database, analyses of OTU clustering and species classification were carried out.

### 2.7. Statistical Analysis

Data were arranged using Microsoft Excel 2013 (Microsoft Corporation, Redmond, WA, USA). Origin 2021 (Origin Software, Inc., Guangzhou, China) was used to draw graphs. Data were analyzed with SPSS 20.0 (Statistics software, IBM, Armonk, NY, USA) using one-way ANOVA. Statistical significance was assessed at $p = 0.05$, and means were separated using Duncan's multiple range test.

## 3. Results

### 3.1. Impact of Aerated Irrigation on the Biomass Accumulation of Grape Seedlings

Aerated irrigation had a significant effect on the biomass of each part of the grape seedlings (Table 1). The root weight, branch weight, leaf weight, and total biomass under aerated treatment (aerated irrigation) were, respectively, 11.20%, 13.59%, 22.95%, and 14.43% higher than those that received non-aerated treatment (CK), demonstrating a significant difference ($p \leq 0.05$). The root–shoot ratio of the seedlings that received aerated treatment was lower than that of the CK seedlings, but the difference was not significant ($p > 0.05$).

**Table 1.** Effects of aerated irrigation on the biomass of 'Red Globe' grape seedlings.

| Treatment | Root Weight (g) | Branch Weight (g) | Leaf Weight (g) | Total Biomass (g) | Root–Shoot Ratio |
|---|---|---|---|---|---|
| Aeration | 59.48 ± 4.23 a | 51.64 ± 6.14 a | 30.05 ± 2.32 a | 141.17 ± 8.76 a | 0728 a |
| No aeration | 53.47 ± 5.85 b | 45.46 ± 3.05 b | 24.44 ± 1.21 b | 123.37 ± 9.91 b | 0.765 a |

Notes: The data presented here are the mean of three replicates ± standard error (SE). Lower-case letters (a,b) within columns denote significant difference at $p \leq 0.05$.

### 3.2. Impact of Aerated Irrigation on the Root Morphology of Grape Seedlings in Different Soil Layers

As shown in Figure 2, aerated treatment significantly changed the distribution and morphology of grape seedling roots in different soil layers. At a soil depth of 0–20 cm, the root length, root surface area, root volume, and number of root tips of the plants that received aerated treatment were significantly lower than those of the CK plants ($p \leq 0.05$). At a soil depth of 20–40 cm, the root length, root surface area, root volume, and number of root tips for the seedlings that received the aerated treatment were, respectively, 38.5%, 32.1%, 62.1%, and 23.4% higher than those of the CK plants, representing significant differences ($p \leq 0.05$). At a soil depth of 20–40 cm, aerated irrigation significantly increased the average root diameter and root volume, but there was no significant difference in the other indexes between the two treatments ($p > 0.05$).

### 3.3. Impact of Aerated Irrigation on Soil Fungal Community Diversity

The high-throughput sequencing results showed that a total of 1,398,883 valid sequences were obtained for all samples. Among these, 60,547 sequences were minimum, and 86,423 sequences were maximum; the average number of sequences was 77,716. The dominant phyla of soil fungi under both treatments were Ascomycota (91.31%) and Basidiomycota (5.55%), and their average relative abundance was greater than 5%. The relative abundances of Glomeromycota and Zygomycota were 0.5% and 0.29%, respectively, with an average relative abundance of less than 1% (Figure S1, Supplementary Materials). Dilution curves (Figure S2, Supplementary Materials) can directly reflect the species richness in the samples. The dilution curves of the six samples tended to level off as the sequencing

amount increased, indicating that the sampling was reasonable and the sequencing data reflected the actual situation of the fungal community in the samples. As shown in Figure 3, the Shannon and Simpson diversity indices for the fungal community of the CK soil were lower than those of the soil that received aerated treatment at different soil depths. The richness indices chao1 and ACE for the fungal community of the soil that received aerated treatment were higher at each soil depth than those for the CK soil, but the difference was insignificant ($p > 0.05$). At a soil depth of 40–60 cm, the ACE index of the CK soil was lower than that at the other depths and significantly lower than that of the aeration-treated soil at a soil depth of 0–20 cm ($p \leq 0.05$).

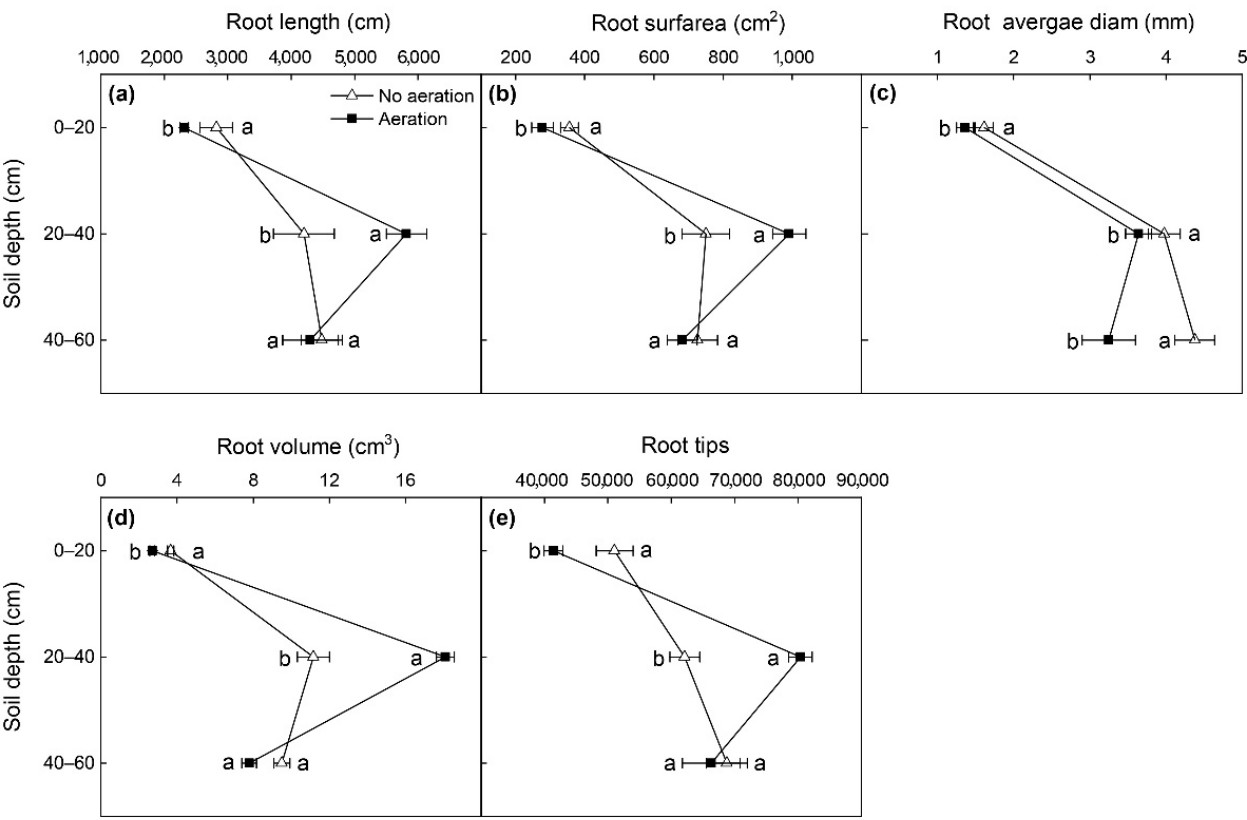

**Figure 2.** (**a**) Root length (cm), (**b**) root surface area (m$^2$), (**c**) mean root diameter (mm), (**d**) root volume (cm$^3$), and (**e**) root tips of grape seedlings at different soil depths under aerated and non-aerated subsurface drip irrigation treatments. Statistical significance was determined by one-way ANOVA (mean $\pm$ SD, $n$ = 3). Data points accompanied by lower-case letters are considered significantly different ($p \leq 0.05$).

*3.4. Impact of Aerated Irrigation on Fungi at Phylum, Class, Order, Family, and Genus Levels*

The relative abundances of dominant fungi (phylum, class, order, family, and genus) at different taxonomic levels are shown in Figure 4. The dominant phyla at each depth were Ascomycota, Basidiomycota, Chytridiomycota, and Glomeromycota (Figure 4a). The dominant classes (Figure 4b) at each depth were Leotiomycetes, Agaricomycetes, Sordariomycetes, Eurotiomycetes, and Glomeromycetes. The dominant orders (Figure 4c) included Hypocreales, Sordariales, Microascales, and Eurotiales. The dominant families (Figure 4d) were Pseudeurotiaceae, Nectriaceae, Chaetomiaceae, Microascaceae, and Trichocomaceae. The analysis of fungi genera with a relative abundance greater than 0.5% showed that the aerated treatment had a significant effect on *Fusarium*, *Microascus*, *Chaetomium*, and *Aspergillus* in soil (Figure 4e). As shown in Figure 5, the relative abundance of *Fusarium* under the aerated treatment was higher than under the CK at all soil layers, especially at a soil depth of 20–40 cm, which was higher by 42.2%—a significant difference ($p \leq 0.05$). The relative abundance of *Microascus* and *Aspergillus* was higher under the aerated treatment

than in CK soil at a soil depth of 0–20 cm, but the difference was insignificant (*p* > 0.05) (Figure 5b,d).

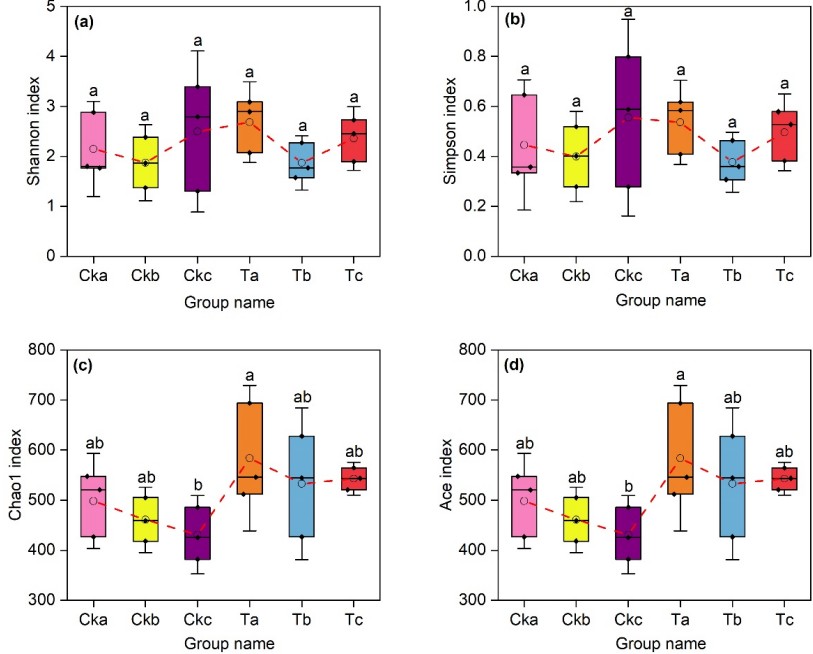

**Figure 3.** Soil fungi. (**a**) Shannon index, (**b**) Simpson index, (**c**) Chao1 index, and (**d**) ACE index at different soil depths under aerated and non-aerated subsurface drip irrigation treatments. The boxes span from the first to the third quartiles; the hollow circles represent the average; the center lines represent the median and the whiskers mean average ± 1.5 SD, *n* = 3. Aerated treatment: Ta, Tb, and Tc represent soil depths of 0–20, 20–40, and 40–50 cm, respectively. No aerated treatment: CKa, CKb, and CKc represent soil depths of 0–20, 20–40, and 40–50 cm, respectively. Data points accompanied by lower-case letters are considered significantly different (*p* ≤ 0.05).

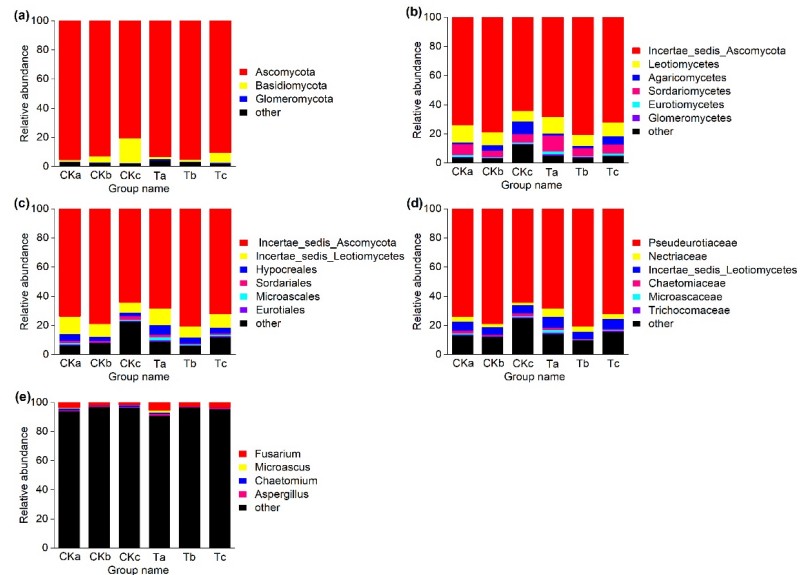

**Figure 4.** Relative abundances of dominant (**a**) fungi phyla (relative abundance ≥ 0.5%), (**b**) fungi class (relative abundance ≥ 0.5%), (**c**) fungi order (relative abundance ≥ 0.5%), (**d**) fungi family (relative abundance ≥ 0.5%), and (**e**) fungi genera (relative abundance ≥ 0.5%) in the soil fungi communities under the two treatments at different soil depths. Aerated treatment: Ta, Tb, and Tc represent soil depths of 0–20, 20–40, and 40–50 cm, respectively. No aerated treatment: CKa, CKb, and CKc represent soil depths of 0–20, 20–40, and 40–50 cm, respectively.

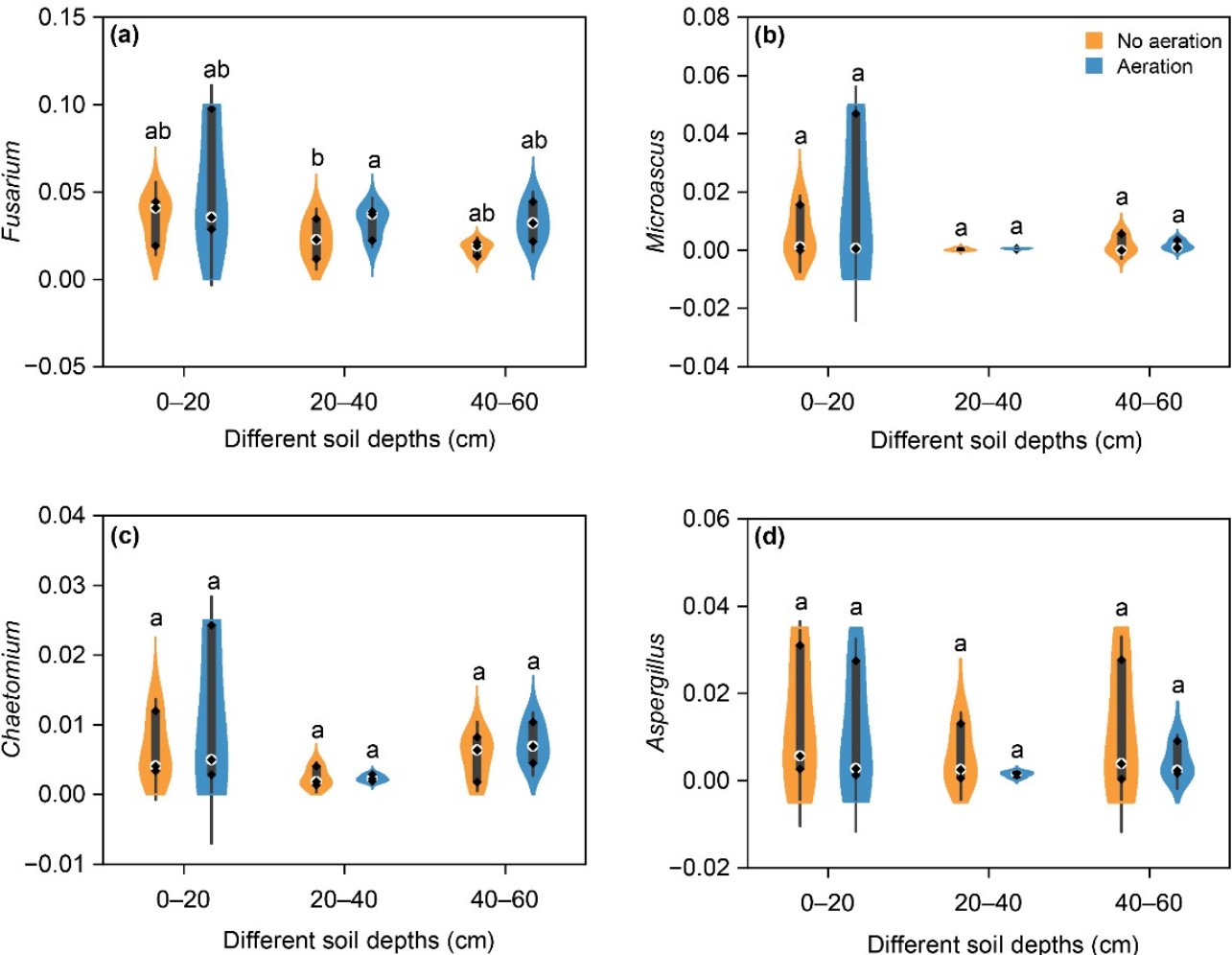

**Figure 5.** Effects of aerated irrigation treatment on the relative abundance of four dominant fungal genera: (**a**) *Fusarium*, (**b**) *Microascus*, (**c**) *Chaetomium*, and (**d**) *Aspergillus*. In the violin-shaped plots, the white dot is the median, the black box ranges between the 25% and 75% quartiles, the black line is the whiskers (mean average $\pm$ 1.5 SD, $n = 3$), and the outer shape is the kernel density estimate. Data points accompanied by lower-case letters are considered significantly different ($p \leq 0.05$).

*3.5. Impact of Aerated Irrigation on Harmful and Beneficial Fungal Genera*

*Alternaria* and *Gibberella* are fungal genera that are harmful to plants. At soil depths of 0–20 and 40–60 cm, the relative abundance of *Alternaria* was lower under the aerated treatment than in the CK soil ($p > 0.05$) (Figure 6a). At each soil layer, the relative abundance of *Gibberella* was higher under the aerated treatment than under the CK, and significantly higher at a soil depth of 20–40 cm (Figure 6b). *Mortierella*, *Cladosporium*, and *Glomus* are beneficial fungal genera. Compared with the CK, the relative abundances of the three fungal genera at 0–20 and 40–60 cm soil depths were higher under the aerated treatment, and the relative abundance of *Mortierella* was 180.6% higher at a soil depth of 0–20 cm ($p \leq 0.05$). At soil depths of 20–40 cm and 40–60 cm, the abundance of both *Mortierella* and *Glomus* was higher under the aerated treatment than under the CK, but the difference was insignificant ($p > 0.05$) (Figure 6c,e).

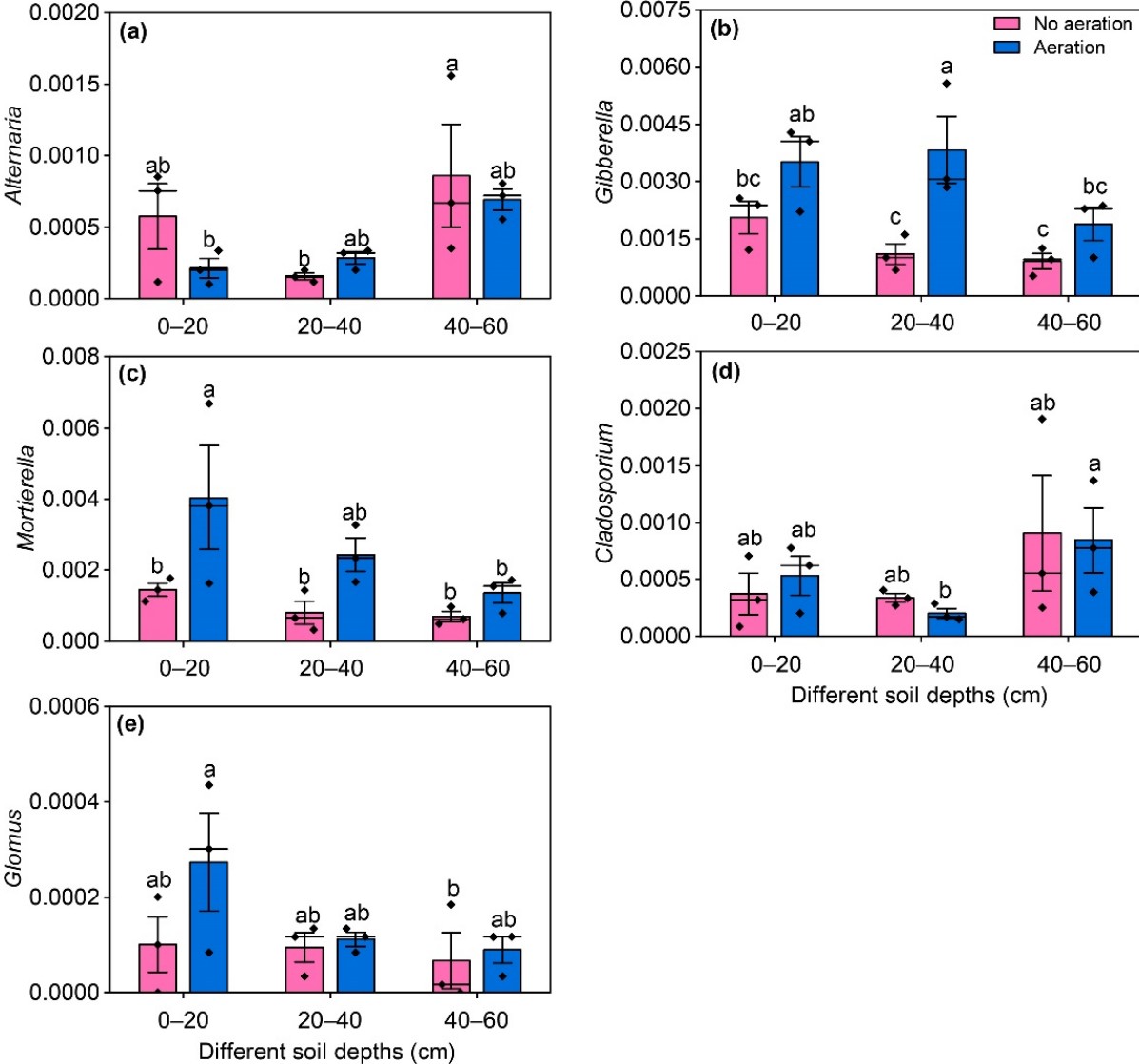

**Figure 6.** Effect of aerated irrigation treatment on the relative abundance of the harmful fungal genera (**a**) *Alternaria* and (**b**) *Gibberella*, and the beneficial fungal genera (**c**) *Mortierella*, (**d**) *Cladosporium*, and (**e**) *Glomus*. The boxes span from the first to the third quartiles. The center lines represent the median, and the whiskers represent the mean average $\pm$ SD, *n* = 3. The data points at the ends of the whiskers represent the outliers. Data points accompanied by lower-case letters are considered significantly different ($p \leq 0.05$).

*3.6. Impact of Aerated Irrigation on the Fungal Community Structure at Different Soil Depths*

Figure 7 illustrates the PCA of changes in the soil fungal community structure under different treatments at different soil depths. It can be observed that the contribution rates of the two principal components were 10.63% and 10.17%. Aerated treatment can be distinguished from CK in the second principal component (PC2), while there was little change in the first principal component (PC1), indicating that aerated treatment can change the structure of the soil fungal community. The top-35 most abundant fungi genera were selected for heat map analysis (Figure 8). Specifically, in Ta, the relative abundances of *Hydropisphaera*, *Microascus*, *Humicoia*, and *Arachnomyces* were high; in Tb, *Malassezia* and *Metarhizium* had high relative abundance; in CKa, *Kernia* and *Leucosphaerina* were high in relative abundance. Meanwhile, in CKc, the five genera with high relative abundance were *Madurella*, *Monodictys*, *Remersonia*, *Mycothermus*, and *Scedosporium*.

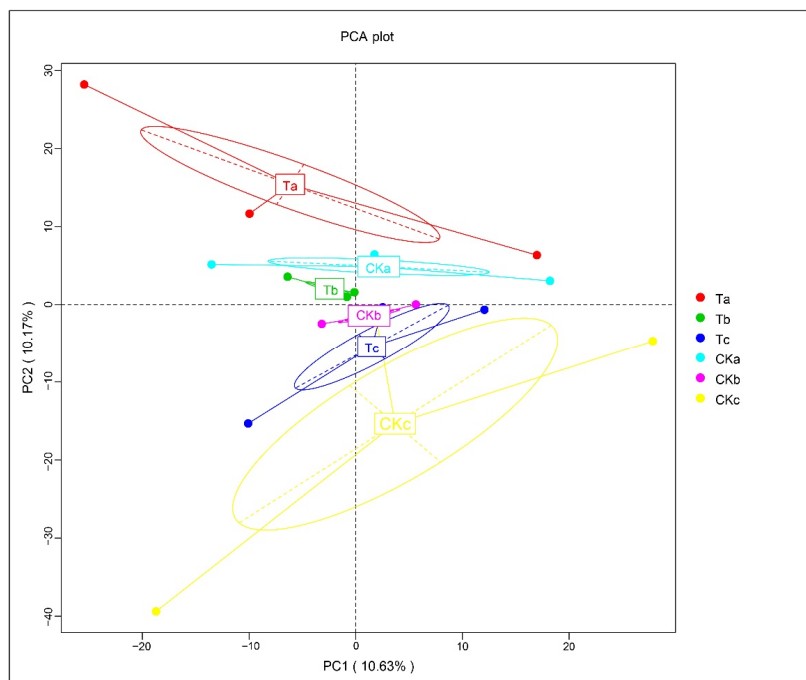

**Figure 7.** Principal component analysis of fungal community structure in soil samples under different treatments. The points with different colors indicate different groups. Aerated treatment: Ta, Tb, and Tc represent soil depths of 0–20, 20–40, and 40–50 cm, respectively. No aerated treatment: CKa, CKb, and CKc represent soil depths of 0–20, 20–40, and 40–50 cm, respectively.

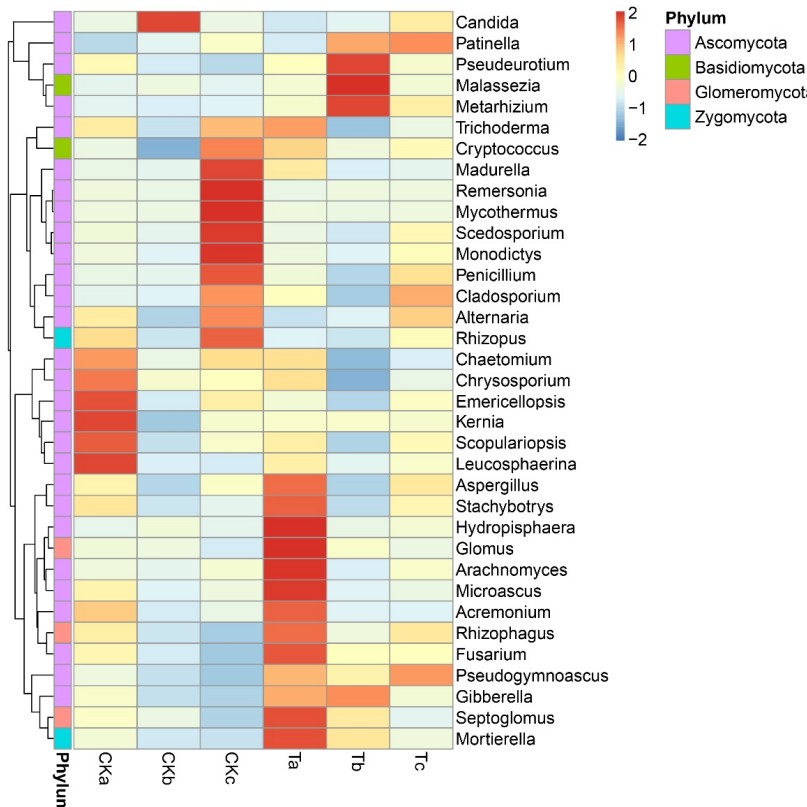

**Figure 8.** The heat map of species clustering at the genus level of fungi in the different soil samples. Red and blue represent the genera with higher abundance and lower abundance, respectively. Aerated treatment: Ta, Tb, and Tc represent soil depths of 0–20, 20–40, and 40–50 cm, respectively. No aerated treatment: CKa, CKb, and CKc represent soil depths of 0–20, 20–40, and 40–50 cm, respectively.

## 4. Discussion

### 4.1. Impact of Aerated Irrigation on the Biomass and Root Distribution of Grape Seedlings

Biomass is an important parameter for evaluating crop growth and development, being intrinsically linked to yield. Changes in biomass can directly determine the ability of crops to accumulate organic matter through net photosynthesis, therefore reflecting the growth and nutritional status of crops [42]. Wen et al. [43] reported that the heights and stem diameters of tomato plants undergoing aerated irrigation increased by 36.54% and 6.81%, respectively, compared with those not undergoing aerated irrigation. Li et al. [44] reported that increasing oxygen levels in the rhizosphere of crops stimulated the canopy growth of crops. For plants, the root system needs sufficient oxygen to meet plant nutrient uptake and water uptake through aerobic respiration to maintain their metabolic needs [45]. Aerated drip irrigation helps to enhance aerobic respiration of the soil in the root zone and improve the respiration efficiency of the roots, which is conducive to mineral nutrient uptake and photosynthetic rate, which in turn promotes plant growth and development [17,46]. The results of this study showed that aerated irrigation had a significant effect on the root weight, branch weight, leaf weight, and total biomass of grape seedlings (Table 1). Compared with the CK, aerated irrigation increased root weight by 11.20%, branch weight by 13.59%, leaf weight by 22.95%, and total biomass by 14.43%, but reduced the root–shoot ratio, which is consistent with the results of the experiment conducted on tomato plants by Zhu et al. [47]. These results indicate that aerated irrigation promotes crop crown growth and increases biomass accumulation by improving the aeration of plant roots, which has a positive impact on plant growth. In summary, the aerated drip irrigation technology can achieve certain economic benefits, the introduction and application in agriculture is meaningful.

Substantial increases both in the number of roots and in root length were observed in crops undergoing aerated drip irrigation; these increases contributed to the efficient uptake of nutrients and water [12,48]. Essah [49] observed that aerated treatment resulted in an increase in root dry weight, tuber yield, and root length in an aerated irrigation test on potato plants. In our experiment, aerated irrigation significantly increased the root length, root surface area, root volume, and the number of root tips of grape seedlings at a soil depth of 20–40 cm. This indicates that the growth of the root system responded positively to aerated irrigation and that aerated irrigation could supply oxygen to the root zone. By injecting air to the rhizosphere of roots, aerated irrigation improves the root respiration environment and promotes the growth of the root system. The increases in root length, root surface area, root volume, and the number of root tips improve the efficiency of the root system in absorbing and transporting water, which is conducive to the growth of the aboveground part of plants [50]. In contrast, at a soil depth of 0–20 cm, aerated irrigation significantly reduced root length, root surface area, root volume, and the number of root tips. At a soil depth of 40–60 cm, there was little difference between the aerated irrigation treatment and the CK, except for the average root diameter and root volume (Figure 2). It may be that the different burial depths of the subsurface drip irrigation units can lead to gas being differently distributed in the soil across different soil layers [51]. Therefore, the higher oxygen concentration in the soil layer near the aeration tank promoted root growth in the grape seedlings' soil layer at a depth of 20–40 cm.

### 4.2. Impact of Aerated Irrigation on the Fungal Community Structure in Grape Rhizosphere Soil

Soil microbial diversity is an important indicator for the health of soil environments [52]. The analysis of the Shannon and Simpson indices of fungal community diversity showed that the aerated treatment reduced the diversity of fungal populations at the depths of 20–40 cm and 40–60 cm in the rhizosphere soil. This indicates that the frequency and intensity of aeration in this experiment had a negative effect on the community diversity of deep soil bacteria; the specific reasons for this need to be further investigated due to the complexity of the soil fungal structure. As the soil depth increased, there was a downward trend in the number of microorganisms [53]. This study's analysis of the fungal community richness indices chao1 and ACE (Figure 3) showed that fungal abundance in the CK

soil declined as the soil depth increased. Additionally, aerated irrigation increased the abundance of soil fungal communities in different soil layers and changed the trend of fungal abundance decreasing with soil depth, a possible reason for which is that aerated irrigation promotes the growth of some dominant fungi. In this case, these fungi increased in quantity.

The number and species of inter-root soil microorganisms are important factors affecting plant growth and development [54,55]. Fungi are one of the crucial components of the community of soil microorganisms and they play a unique driving role in both the circulation of biochemical substances in the soil and the monitoring of soil-borne diseases [56]. In this study, the dominant phyla were Ascomycota and Basidiomycota, with relative abundances accounting for 91.31% and 5.55%, respectively (Figure S1). The abundance of both Ascomycota and Basidiomicota responded positively to changes in soil pH, with Ascomycota being more abundant in soils with higher pH and Basidiomicota being more abundant in soils with lower pH [57]. The soil in this experiment was alkaline, which may be the reason for the high abundance of ascomycetes and low relative abundance of Basidiomicota. Aerated irrigation introduces air into the soil and affects the emission and distribution of $CO_2$, $NO_2$, and other gases in the soil, thus affecting the respiration of rhizosphere microorganisms and consequently the number of soil Ascomycetes and Basidiomycetes phyla. The dominant fungi genera included *Microascus*, *Chaetomium*, and *Aspergillus*, as well as *Fusarium*—a harmful fungal genus that can cause root rot in grapes. Changes in the levels of oxygen dissolved in the soil can directly affect the soil's microbial biomass and community structure [58]. In this study, the relative abundance of *Fusarium* in the soil that received aerated treatment was higher than that in the CK soil in all soil layers, and by 42.2% at 20–40 cm. *Fusarium* is widely distributed in deep soils, and it has strong adaptability to new environments. Aerated irrigation improved its living environment, which increased its relative abundance in soil. Aerated irrigation also had an impact on the abundance of the other three dominant fungal genera, but the difference was not significant (Figure 5). According to the heat map of the top 35 fungal genera in terms of relative abundance (Figure 8), the aerated treatment did not have genera in common with the CK soil, indicating that aerated irrigation affected the abundance of the soil fungal community.

Fungi can be divided into three categories according to their functions: saprophytic fungi, pathogenic fungi, and mycorrhizal symbiotic fungi [59]. Soil saprophytic fungi use carbon from above-ground apoplankton and roots to facilitate their metabolism, which is important for the natural recycling of elements such as carbon [60]. *Alternaria* fungi are a variety of plant pathogens. Their main pathogenic mechanism is to cause pathological reactions and damage to infected plant tissues by producing a variety of pathogenic toxins [61,62]. In this study, the relative abundance of *Alternaria* fungi in soil that underwent aerated irrigation was lower than that in the CK soil at depths of 0–20 and 40–60 cm (Figure 6). The test results demonstrate that aerated irrigation restricted the development of *Mortierella*, *Cladosporium*, and *Glomus*, all of which are fungal genera that are beneficial to plants. Some specific *Mortierella* make important contributions to soil nutrient transformation and availability. For example, Ning et al. [63] found that a *Mortierella elongata SX* played a crucial role in nutrient transformation and plant growth promotion in mineral soils. *Mortierella* sp. is capable of dissolving soil phosphorus by releasing multiple organic acids in different soils [64]. A recent study conducted by Tamayo-Vélez and Osorio [65] reported that the addition of *Mortierella* sp. to orchard soil significantly increased the content of available phosphorus, potassium, calcium, magnesium, and boron. *Mortierella* sp. interacts with arbuscular mycorrhizal fungi in saline–alkali soil, which can enhance soil phosphatase activity and promote plant growth [66]. Glomus fungi are mycorrhizal symbiotic fungi that can form mycorrhizae with some plant roots, and mycorrhizal fungi are key components of a sustainable soil–plant system [67]. In this study, the relative abundance of beneficial fungal genera was higher in all soil depths under the aerated treatment than in the without aerated treatment, except for *Cladosporium* at a soil depth of 20–40 cm. Moreover, at a soil depth of 0–20 cm, the relative abundance of *Mortierella* under the aerated treatment was

180.6% higher than that under CK, representing a significant difference ($p \leq 0.05$). The test results demonstrate that aerated irrigation restricted the development of *Alternaria* while partly promoting the growth of the beneficial fungi *Mortierella*, *Cladosporium*, and *Glomus*. In addition, aerated irrigation altered the relative abundance of some beneficial and harmful fungi by injecting air into the rhizosphere soil of plants, which increased the beneficial fungi in the rhizosphere soil, thereby improving the growth environment of plant roots.

**5. Conclusions**

This study demonstrated that aerated irrigation can significantly increase root weight, branch weight, leaf weight, and total biomass ($p \leq 0.05$). Furthermore, aerated irrigation can significantly increase root length, root surface area, root volume, and the number of root tips of grape seedlings at soil depths of 0–20 cm and 20–40 cm ($p \leq 0.05$). In addition, aerated irrigation can effectively increase the abundance of fungi in the soil at various depths, but it has little effect on fungal diversity. Specifically, it increased the abundance of beneficial fungal genera *Mortierella*, *Cladosporium*, and *Glomus*, and reduced the number of pathogenic fungal genera of *Alternaria*. Overall, aerated irrigation represents a promising approach for promoting grape biomass accumulation and root growth and for increasing the abundance of some beneficial fungi. In the future, more attention could be paid to the effects of different aeration cycles, aeration frequency, and aeration intensity on grape growth and rhizosphere microbial communities.

**Supplementary Materials:** The following supporting information can be downloaded at https://www.mdpi.com/article/10.3390/su141912719/s1, Figure S1: Fungal community structure in rhizosphere soil. Figure S2: The fungi dilution curve in rhizosphere soil.

**Author Contributions:** Conceptualization, F.Z. and K.Y.; methodology, F.Z. and K.Y; formal analysis, H.Z. and J.X.; investigation, F.Z. and K.Y.; resources, F.Z., Q.L., J.W., and K.Y.; data curation, H.Z. and J.X.; writing—original draft preparation, H.Z., Q.L., J.W., and J.X.; writing—review and editing, H.Z., J.X., and F.Z; visualization, H.Z., J.X., and F.Z.; supervision, F.Z. and K.Y.; project administration, K.Y. All authors have read and agreed to the published version of the manuscript.

**Funding:** This research was funded by the Natural Science Foundation of China (31760550), the Agricultural Science and Technology Tackling Project in Shihezi City of Bashi (2017HZ05), Shihezi University Youth Innovative Talent Cultivation Program project (CXPY201914), and the Transformation Project of Scientific and Technological Achievements of the Xinjiang Production and Construction Corps (2020BA006).

**Data Availability Statement:** Data are available from the authors upon request.

**Conflicts of Interest:** The authors declare no conflict of interest.

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
