# Peer review of "Effect of Aerated Irrigation on the Growth and Rhizosphere Soil Fungal Community Structure of Greenhouse Grape Seedlings"

_sustainability, doi:10.3390/su141912719_

Round 1
Reviewer 1 Report
General comment:
In this study, aerated irrigation equipment was used to investigate the effects of aerated irrigation on the biomass accumulation, root growth, and soil fungal community structure of grape seedlings. Findings suggest that aerated irrigation is a promising strategy for the promotion of grape root growth and biomass accumulation, and it can also increase the abundance of some beneficial fungi. It’s suggested that more detailed data should be introduced to make some viewpoints more convincing through the manuscript.
Specific comments:
1. Abstract: The detailed critical data should be included, e.g., root length increase by ...%.
2. Line 29: The disadvantages of the use of underground drip irrigation technology are pointed out, such as: deterioration of soil physical and chemical properties, affecting crop growth, yield reduction, etc. Highlight the advantages of aeration irrigation.
3. Line 44: It is recommended to supplement relevant references.
4. Line 46: What conclusions are drawn from the content of this reference study.
5. Line 48: Crop yields and qualities are more convincing by giving specific values.
6. Line 82: Aeration irrigation has done less research on grape growth and development, and suggests why this study was conducted.
7. Line 158: Whether the thermal cycle used for PCR amplification should be indicated.
8. Line 309: It is recommended to increase the aeration of irrigation to promote the cause of plant growth. The aeration treatment promotes gas exchange in the soil in the root zone and increases soil respiration.
9. Line 364: The reasons for the high relative abundance of Ascomycota and the low relative abundance of Basidiomicota are identified.
10. Line 364: It is recommended to describe the relationship between aeration and Ascomycetes or Basidiomycetes.
11. Line 378: It is recommended to indicate the different effects of different fungal communities on plants in rhizosphere soils.
12. Line 417: It is recommended that further research can be conducted to investigate the aeration cycle, aeration frequency, and aeration intensity.
Author Response
- Abstract: The detailed critical data should be included, e.g., root length increase by ...%.
Answer: Thank you for your insightful suggestion. We have modified this sentence to “The results show that aerated irrigation significantly increased the root length, root surface area, root volume, and number of root tips by 38.5%, 32.1%, 62.1%, and 23.4%, respectively, at a soil depth of 20–40 cm (p ≤ 0.05)”. (Abstract, Lines 16 and 19)
- Line 29: The disadvantages of the use of underground drip irrigation technology are pointed out, such as: deterioration of soil physical and chemical properties, affecting crop growth, yield reduction, etc. Highlight the advantages of aeration irrigation.
Answer: Thank you for your insightful suggestion. We added the disadvantages of subsurface drip irrigation and the advantages of aerated irrigation. We have revised the first paragraph of the introduction as follows:
Subsurface drip irrigation is a widely used irrigation technology. Long-term sub-surface drip irrigation has an impact on the soil’s structure and hydraulic properties near the drippers, as well as limiting oxygen diffusion in the root zone of crops, thereby affecting the aerobic respiration of plant roots [1]. Such traditional irrigation techniques drive out the gas in the spaces between soil pores, thus reducing the soil air content between plant roots and causing low oxygen stress in plants [2,3]. Hypoxic stress is one of the important adversity factors affecting normal plant growth and development [4], which increases the reduction of toxic substances in the soil, imbalances the uptake of mineral elements, and disrupts the hormone metabolism in plants [5,6]. Aerated drip irrigation is a new type of irrigation that has emerged in recent years as a refinement and improvement of subsurface drip irrigation. Previous studies have shown that this irrigation method is an innovative water-saving irrigation technology with great development potential [7–9]. Based on traditional subsurface drip irrigation water-saving technology, aerated irrigation uses aeration equipment to dissolve air or other gases into irrigation water; once the water has been mixed with gas, it is then transported to the root zone of the crop through a drip irrigation pipeline [10]. Aerated irrigation can increase soil oxygen content and porosity, alleviate oxygen shortages during irrigation, and promote soil microbial and root respiration, improving root traits and biomass [11–14]. Aerated irrigation can improve the ratio of solid, liquid and gas phases in the soil, which can effectively regulate soil microbial activity, the effectiveness of soil nutrients and soil redox reactions and other soil microenvironments [15]. In turn, it improves soil properties and increases soil productivity, thus enhancing root uptake and utilization of soil nutrients and water [16,17]. (Introduction, Lines 32 and 53)
Reference
- Friedman, S.P.; Naftaliev, B. A Survey of the aeration status of drip-irrigated orchards. Agric. Water Manag. 2012, 115, 132–147. DOI: 10.1016/j.agwat.2012.08.015
- Fiebig, A.; Dodd, I.C. Inhibition of tomato shoot growth by over-irrigation is linked to nitrogen deficiency and ethylene. Physiol. Plant. 2015, 156, 70–83. DOI: 10.1111/ppl.12343
- Patel, M.K; Pandey, S.; Burritt, D.J., Lam-Son, P.T. Plant responses to low-oxygen stress: Interplay between ROS and NO signaling pathways. Environ. Exp. Bot. 2019, 161,134–142. DOI: 10.1016/j.envexpbot.2019.02.013
- Niu, W.Q.; Guo, Q.; Zhou, X.B.; Helmers, M.J. Effect of aeration and soil water redistribution on the air permeability under subsurface drip irrigation. Soil Sci. Soc. Am. J. 2012, 76, 815–820. DOI: 10.2136/sssaj2011.0329
- Licausi, F.; Giuntoli, B.; Perata, P. Similar and yet different: oxygen sensing in animals and plants. Trends Plant Sci. 2020, 25, 6–9. DOI: 10.1016/j.tplants.2019.10.013
- Kläring, H.P.; Zude, M. Sensing of tomato plant response to hypoxia in the root environment. Sci. Hortic. 2009, 122, 17–25. DOI: 10.1016/j.scienta.2009.03.029
- Abuarab, M.E.; El-Mogy, M.M.; Hassan, A.M.; Abdeldaym, E.A.; Abdelkader, N.H.; B.I. El-Sawy, M. The effects of root aeration and different soil conditioners on the nutritional values, yield, and water productivity of potato in clay loam soil. Agronomy 2019, 9, 418. DOI: 10.3390/agronomy9080418
- Yang, X.; Fan, J.; Ge, J.; Luo, Z. Effect of irrigation with activated water on root morphology of hydroponic rice and wheat seedlings. Agronomy 2022, 12, 1068. DOI: 10.3390/agronomy12051068
- Yu, Z.Z.; Wang, C.; Zou, H.F.; Wang, H.X.; Li, H.L.; Sun, H.T.; Yu, D.H. The effects of aerated irrigation on soil respiration and the yield of the maize root zone. Sustainability-Basel 2022, 14, 4378. DOI: 10.3390/su14084378
- Cao, X.S.; Li, H.P.; Zheng, H.X.; Feng, Y.Y.; Cheng, Z.Z.; Zhao, Q.H. Effects of aerated irrigation on soil fertility and crop growth in root zone. Agric. Res. Arid. Areas 2020, 38, 183–189. DOI: 10.7606/j.issn.1000-7601.2020.01.24
- Heuberger, H.; Livet, J.; Schnitzler, W. Effect of soil aeration on nitrogen availability and growth of selected vegetables–preliminary results. Acta. Hortic. 2001, 56, 147–154. DOI: 10.17660/ActaHortic.2001.563.18.
- Bhattarai, S.P.; Su, N.; Midmore, D.J. Oxygation unlocks yield potentials of crops in oxygen limited soil environments. Adv. Agron. 2005, 88, 313–377. DOI: 10.1016/S0065-2113(05)88008-3
- Du, Y.D.; Niu, W.Q.; Gu, X.B.; Zhang, Q.; Cui, B.J.; Zhao, Y. Crop yield and water use efficiency under aerated irrigation: A meta-analysis. Agric. Water Manag. 2018, 210, 158–164. DOI: 10.1016/j.agwat.2018.07.038
- Niu, W.Q.; Zang, X.; Jia, Z.X.; Shao, H.B. Effects of rhizosphere ventilation on soil enzyme activities of potted tomato under different soil water stress. CLEN–soil, Air, Water 2021, 40, 225–232. DOI: 10.1002/clen.201100
- Yu, Z.Z.; Wang, C.; Zou, H.F.; Wang, H.X.; Li, H.L.; Sun, H.T.; Yu, D.S. The Effects of Aerated Irrigation on Soil Respiration and the Yield of the Maize Root Zone. Sustainability 2022, 14, 1–18.
- Zhang, Q.; Du U.D; Cui, B.J.; Sun, J.; Wang, J.; Wu, M.L.; Niu, W.Q. Aerated irrigation offsets the negative effects of nitrogen reduction on crop growth and water-nitrogen utilization. J. Clean. Prod. 2021, 313, 127917. DOI: 10.1016/j.jclepro.2021.127917
- Cui, B.J.; Niu, W.Q.; Du, Y.D.; Zhang, Q. Response of yield and nitrogen use efficiency to aerated irrigation and N application rate in greenhouse cucumber. Scientia Horticulturae, 2020, 265:109220. DOI: 10.1016/j.scienta.2020.1092203. Line 44: It is recommended to supplement relevant references.
- Line 44: It is recommended to supplement relevant references.
Answer: Thank you for your insightful suggestion. We have added three references and changed the sentence in this section. Aerated irrigation can increase soil oxygen content and porosity, alleviate oxygen shortages during irrigation, and promote soil microbial and root respiration, improving root traits and biomass. [11–14]. (Introduction, Lines 47 and 49)
Reference
- Heuberger, H.; Livet, J.; Schnitzler, W. Effect of soil aeration on nitrogen availability and growth of selected vegetables–preliminary results. Acta. Hortic. 2001, 56, 147–154. DOI: 10.17660/ActaHortic.2001.563.18.
- Bhattarai, S.P.; Su, N.; Midmore, D.J. Oxygation unlocks yield potentials of crops in oxygen limited soil environments. Adv. Agron. 2005, 88, 313–377. DOI: 10.1016/S0065-2113(05)88008-3
- Du, Y.D.; Niu, W.Q.; Gu, X.B.; Zhang, Q.; Cui, B.J.; Zhao, Y. Crop yield and water use efficiency under aerated irrigation: A meta-analysis. Agric. Water Manag. 2018, 210, 158–164. DOI: 10.1016/j.agwat.2018.07.038
- Niu, W.Q.; Zang, X.; Jia, Z.X.; Shao, H.B. Effects of rhizosphere ventilation on soil enzyme activities of potted tomato under different soil water stress. CLEN–soil, Air, Water 2021, 40, 225–232. DOI: 10.1002/clen.201100
- Line 46: What conclusions are drawn from the content of this reference study.
Answer: Thank you for your insightful suggestion. We have modified this sentence to “Previous studies on aerated irrigation have shown that aerated irrigation can promote plant growth, increase crop yield and improve fruit quality [18–20]”. (Introduction, Lines 54 and 55)
- Line 48: Crop yields and qualities are more convincing by giving specific values.
Answer: Thank you for your insightful suggestion. I have modified this sentence to “Yuan et al. [21] reported that increasing oxygen levels in the root zones of crops had a positive impact, stimulating above-ground growth and increasing tomato yield increased by 8% on average”. (Introduction, Lines 56 and 58)
- Line 82: Aeration irrigation has done less research on grape growth and development, and suggests why this study was conducted.
Answer: Thank you for your insightful suggestion. We have added “However, there is a lack of systematic studies on the effects of aeration irrigation on grape growth and development and rhizosphere soil fungal community”. (Introduction, Lines 92 and 94)
- Line 158: Whether the thermal cycle used for PCR amplification should be indicated.
Answer: Thank you for your insightful suggestion. We have added “PCR was performed under the following conditions: 3 min of denaturation at 95 ℃, followed by 30 cycles, 30 s of denaturation at 95 ℃, 30 s of annealing at 55 ℃, 30 s of extension at 72 ℃, and a final extension step at 72 ℃ for 5 min”. (Materials and Methods, Lines 177 and 180)
- Line 309: It is recommended to increase the aeration of irrigation to promote the cause of plant growth. The aeration treatment promotes gas exchange in the soil in the root zone and increases soil respiration.
Answer: Thank you for your insightful suggestion. We have added “For plants, the root system needs sufficient oxygen to meet plant nutrient uptake and water uptake through aerobic respiration to maintain their metabolic needs [46]. Aerated drip irrigation helps to enhance aerobic respiration of the soil in the root zone and improve the respiration efficiency of the roots, which is conducive to mineral nutrient uptake and photosynthetic rate, which in turn promotes plant growth and development [47,48]”. (Discussion, Lines 338 and 343)
Reference
- Mustroph A, Albrecht G. Tolerance of crop plants to oxygen deficiency stress: Fermenting activity and photosynthetic ca-pacity of entire seedlings under hypoxia and anomia. Physiol. Plantarum 2003, 117, 508–520. DOI: 10.1034/j.1399-3054.2003.00051.x
- Zhu, Y.; Dyck, M.; Cai, H.J.; Song, L.B.; Chen, H. The effects of aerated irrigation on soil respiration, oxygen, and porosity. J. Integr. Agr. 2019, 18, 2854–2868. DOI: 10.1016/S2095-3119(19)62618-3.
- Zhu, J.J.; Xu, N.; Siddique, K.H.M.; Zhang, Z.H.; Niu, W.Q. Aerated drip irrigation improves water and nitrogen uptake efficiencies of tomato roots with associated changes in the antioxidant system. Sci. Hortic. 2022, 306,111471. DOI: 10.1016/j.scienta.2022.111471.
- Line 364: The reasons for the high relative abundance of Ascomycota and the low relative abundance of Basidiomicota are identified.
Answer: Thank you for your insightful suggestion. We have added “The abundance of both Ascomycota and Basidiomicota responded positively to changes in soil pH, with Ascomycota being more abundant in soils with higher pH and Basidiomicota being more abundant in soils with lower pH [60]. The soil in this experiment was alkaline, which may be the reason for the high abundance of ascomycetes and low relative abundance of Basidiomicota”. (Discussion, Lines 393 and 398)
Reference
- Tedersoo, L.; Bahram, M.; Polme, S.; Koljalg, U.; Yorou, N. S.; Wijesundera, R.; Ruiz, L. V.; Vasco-Palacios, A. M.; Thu, P. Q.; Suija, A.; Smith, M. E.; Sharp, C.; Saluveer, E.; Saitta, A.; Rosas, M.; Riit, T.; Ratkowsky, D.; Pritsch, K.; Poldmaa, K.; Piepenbring, M.; Phosri, C.; Peterson, M.; Parts, K.; Partel, K.; Otsing, E.; Nouhra, E.; Njouonkou, A.L.; Nilsson, R.H.; Morgado, L. N. ; Mayor, J.; May, T. W.; Majuakim, L.; Lodge, D. J.; Lee, S. S.; Larsson, K. H.; Kohout, P.; Hosaka, K.; Hiiesalu, I.; Henkel, T. W.; Harend, H.; Guo, L. D.; Greslebin, A.; Grelet, G.; Geml, J.; Gates, G.; Dunstan, W.; Dunk, C.; Drenkhan, R.; Dearnaley, J.; De Kesel, A.; Dang, T.; Chen, X.; Buegger, F.; Brearley, F. Q.; Bonito, G.; Anslan, S.; Abell, S.; Abarenkov, K. Global diversity and geography of soil fungi. Science 2014, 346, 1078–+. DOI: 10.1126/science.1256688.
- Line 364: It is recommended to describe the relationship between aeration and Ascomycetes or Basidiomycetes.
Answer: Thank you for your insightful suggestion. We have added “Aerated irrigation introduces air into the soil and affects the emission and distribution of CO2, NO2 and other gases in the soil, thus affecting the respiration of rhizosphere microorganisms and consequently the number of soil Ascomycetes and Basidiomycetes phyla”. (Discussion, Lines 398 and 401)
- Line 378: It is recommended to indicate the different effects of different fungal communities on plants in rhizosphere soils.
Answer: Thank you for your insightful suggestion. We have added the effects of soil saprophytic fungi on plants and the effects of pathogenic and commensal fungi on plants in inter-rhizosphere soils mentioned in the article in lines 418-419 and 432-434 of the discussion, respectively. We have added “Soil saprophytic fungi use carbon from above-ground apoplankton and roots to facilitate their metabolism, which is important for the natural recycling of elements such as carbon [63]”. (Discussion, Lines 415 and 417)
Reference
- Pinzari, F.; Ceci. A.; Samra, N.A.; Canfora, L.; Maggi, O.; Persiani, A. Phenotype MicroArray system in the study of fungal functional diversity and catabolic versatility. Research in Microbiology 2016, 167, 710–722. DOI: 10.1016/j.resmic.2016.05.008
- Line 417: It is recommended that further research can be conducted to investigate the aeration cycle, aeration frequency, and aeration intensity.
Answer: Thank you for your insightful suggestion. We have added some future research directions and extensions of my research in the conclusion. We have added “In the future, more attention could be paid to the effects of different aeration cycles, aeration frequency and aeration intensity on grape growth and rhizosphere microbial communities”. (Conclusion, Lines 455 and 457)

Reviewer 2 Report
The study reported the effect of aerated irrigation on the growth and rhizosphere soil fungal community structure of greenhouse grape seedlings. The title and the aim are important and presented wealth of potentially interesting data. A lot of works and revisions have been done already. The manuscript certainly deserves publication in the journal in the current form.
Minor comments
L93-94. How was it possible to maintain the relative humidity or do the author want to say that 60-80% relative humidity was reported? Revise intention
L273 and other applicable sections in the entire manuscript. It is aerated not aeration treatment. Revise adjective
Author Response
L93-94. How was it possible to maintain the relative humidity or do the author want to say that 60-80% relative humidity was reported? Revise intention
Answer: The relative humidity of 60%–80% in the greenhouse is based on the statistics of the thermohygrometer measurement records, not on the reports of others.
Thank you for your insightful suggestion. We have modified this sentence to “During the test, the temperature of the greenhouse was 17 °C–33 °C and the relative humidity was 60%–80%”. (Materials and Methods, Lines 104 and 105)
L273 and other applicable sections in the entire manuscript. It is aerated not aeration treatment. Revise adjective
Answer: We are sorry for making this mistake. We have changed the word “aeration treatment” to “aerated treatment” in 39 places throughout the manuscript.
Author Response
Answer: We mentioned in the introduction that aerated irrigation has been used on fruit trees with a positive impact, and in our research on grapes, we also obtained that aerated irrigation can promote the growth and development of grapes, so we think this technology is worth applying. Thank you for your insightful suggestion. I added content about the application. We have added “In summary, the aerated drip irrigation technology can achieve certain economic benefits, the introduction and application in agriculture is meaningful”. (Discussion, Lines 350 and 352)
Round 2
Reviewer 1 Report
The authors have well addressed all the issues, I have no more comments.
Author Response
Thanks to your valuable comments on this manuscript, the quality of our manuscript has improved greatly. Thanks again for all your hard work!